# Fully-Automated Multi-View Classification for Lesion Interpretation in Mammography

**Diogo Lourenço Cortez**                                    DIOGO.CORTEZ@TECNICO.ULISBOA.PT
**Carlos Santiago**                                    CARLOS.SANTIAGO@TECNICO.ULISBOA.PT
**Jacinto C. Nascimento**                                    JAN@ISR.TECNICO.ULISBOA.PT
*Department of Computer Science and Engineering,*
*Insituto Superior Técnico, Av. Rovisco Pais 1, 1049-001 Lisbon, Portugal*

**Editors:** Under Review for MIDL 2021

## Abstract

Given the extreme importance in early detection of breast cancer, a compelling search for Computer-Aided Detection (CAD) techniques drove Deep Learning (DL) researchers to investigate potential mammography screening applications. This work proposes the use of sophisticated and recently proposed Convolutional Neural Networks (CNNs) for classification and segmentation of mass and Micro-Calcification (MC) lesions on the INbreast dataset via a novel fully-automated pipeline. For segmentation, an Attention Dense U-Net model is used to provide segmentations for masses for MCs. The classification stage is performed via a Dense Multi-View model, receiving an enriched input with the previous predicted segmentations, achieving performance on par with state of the art for fully-automated classification of breast screening exams (Normal, Benign, Malignant), achieving a 3-Class Mean AUC of $(0.79 \pm 0.06)$.

**Keywords:** Deep learning, Medical Imaging, Multi-View Models, Convolutional Neural Networks, Transfer Learning, Image Segmentation.

## 1. Introduction

Breast cancer is the most frequent cancer among women (Ferlay et al., 2018; Siegel et al., 2019). However, declining breast cancer mortality trends have been reported due to the combined effects of earlier detection and treatment (Smith et al., 2006).

Mammography is the imaging modality that contributed the most in reducing breast cancer mortality (Kopans, 2002). Recent studies indicate that DL driven CAD systems can improve the performance of radiologists in exam evaluation (Wu et al., 2019). There are two distinct tasks in which these CAD systems are helpful: classification and segmentation.

Regarding classification, Multi-View assessment of Craniocaudal (CC) and Mediolateral Oblique (MLO) views has been shown to lead to better performance (Carneiro et al., 2017; Wu et al., 2019). Regarding segmentation, recent models are based on encoder-decoder architectures, of which the U-Net is an example (Ronneberger et al., 2015). Following breakthroughs in language translation (Vaswani et al., 2017), attention mechanisms have established a dominant position in encoder-decoder mappings in medical imaging.

Despite of the promise of DL driven CAD systems, recent prominent work in the field makes use of relatively simple models (Carneiro et al., 2017) or require abundant private datasets to train (Wu et al., 2019). In this sense, we propose the development of a fully-automated pipeline, based on sophisticated holistic models, capable of segmenting lesions for an enriched interpretation of lesions in the INbreast dataset (Moreira et al., 2012).

Regarding the segmentation of mammograms, this work contributes by extending the work in (Li et al., 2019) by evaluating the performance of an adapted version of the Attention Dense-U-Net to segment mass and MC lesions (see Section 2.1). Regarding classification, the applicability of dense layers (Huang et al., 2016) was explored in the context of segmented lesion interpretation (see Section 2.2). This novel combination constitutes a major contribution, proposing a DL approach based on holistic models (i.e. single-stage classification and segmentation) capable of producing informative lesion segmentations useful for state of the art fully-automated mammography classification. This is more appealing than the approach in (Carneiro et al., 2017), which made use of dedicated cascaded pipelines to extract masses and MCs (Dhungel et al., 2015; Lu et al., 2016).

## 2. Proposal

The focus of this work lies in the development of a fully-automated pipeline (see Figure 1) for classification and segmentation of potential lesions in Multi-View mammography images. The dataset can be represented as $\mathcal{D} = \{(\boldsymbol{x}^{(p,b)}, \boldsymbol{c}^{(p,b)}, \boldsymbol{m}^{(p,b)}, \boldsymbol{y}^{(p,b)})\}_{p\in\{1,...,P\},b\in\{left,right\}}$, where $\boldsymbol{x} = \{\boldsymbol{x}_{CC}, \boldsymbol{x}_{MLO}\}$ denotes the CC and MLO mammography views, with $\boldsymbol{x}_{CC}, \boldsymbol{x}_{MLO} : \Omega \mapsto \mathbb{R}$ ($\Omega$ represents the image lattice), $\boldsymbol{m} = \{\boldsymbol{m}_{CC}, \boldsymbol{m}_{MLO}\}$ and $\boldsymbol{c} = \{\boldsymbol{c}_{CC}, \boldsymbol{c}_{MLO}\}$ denote the binary mass and MC segmentation maps per view, with $\boldsymbol{m}_{CC}, \boldsymbol{m}_{MLO}, \boldsymbol{c}_{CC}, \boldsymbol{c}_{MLO} : \Omega \mapsto \{0, 1\}$, $\boldsymbol{y} \in \mathcal{Y} = \{0, 1\}^3$ represents the labels, $p \in \{1, ..., P\}$ indexes the patients, and $b \in \{left, right\}$ the indexes the left and right breasts of the patient.

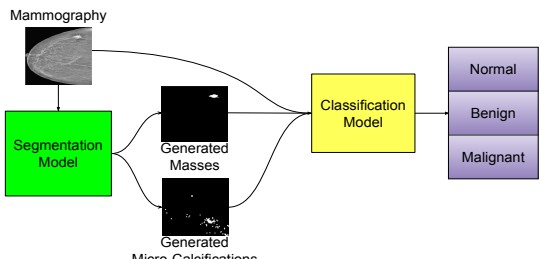

Figure 1: Proposed Fully-Automated Pipeline.

### 2.1. Segmentation Model

For our segmentation model, we adapted the Attention Dense-U-Net architecture (Li et al., 2019) for Single-View mass and MC segmentation (producing two separately trained models). The overall architecture is reminiscent of the typical U-Net, while staying true to the DenseNet-121-BC core (gray-colored in Figure 2(a)) in the encoder part of the network and employing attention in the decoder blocks (red-colored in Figure 2(a)). Because of the aggressive downsize of the first convolution and pooling in the dense core, a notable additional convolutional path from the input is found in the last decoder skip connection (see the top row of Figure 2(a)). Additionally, the filter size is reduced after each decoder block by a factor of 2, inspired by the reduction of filter size seen in DenseNet-121-BC.

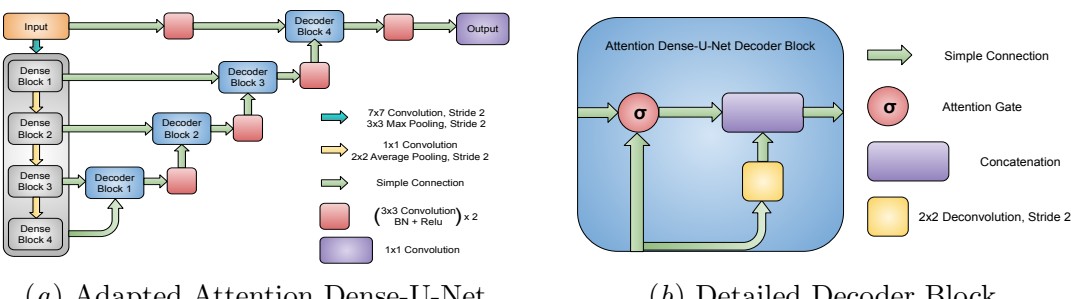

(a) Adapted Attention Dense-U-Net      (b) Detailed Decoder Block

Figure 2: Segmentation Model Architecture.

In our context, the segmentation models can be formally represented by $\mathcal{F}_S^l : \mathcal{X}_S \mapsto \mathcal{Y}_S^l$, where $l \in \{Mass, MC\}$ denotes the lesion type being segmented, $\mathcal{X}_S = \{\boldsymbol{x}_{CC}, \boldsymbol{x}_{MLO}\}$ the mammography scans, $\mathcal{Y}_S^{Mass} = \{\boldsymbol{ma}_{CC}, \boldsymbol{ma}_{MLO}\}$ and $\mathcal{Y}_S^{MC} = \{\boldsymbol{mc}_{CC}, \boldsymbol{mc}_{MLO}\}$ the resulting mass and MC segmentations, with $\boldsymbol{ma}_{CC}, \boldsymbol{ma}_{MLO}, \boldsymbol{mc}_{CC}, \boldsymbol{mc}_{MLO} : \Omega \mapsto [0,1]$.

## 2.2. Classification Model

The proposed Multi-View classification model is depicted in Figure 2. For each exam, a dual path processes each view, enriched with the corresponding segmented output masses and MCs from the proposed segmentation model. This way, the informative features are automatically processed and selected for an enhanced assessment of lesion interpretation.

The model can then be formally represented by $\mathcal{F}_C : (\mathcal{X}_C^{CC}, \mathcal{X}_C^{MLO}) \mapsto \mathcal{Y}_C$, where $\mathcal{X}_C^{CC} = \{[\boldsymbol{x}_{CC}, \boldsymbol{ma}_{CC}, \boldsymbol{mc}_{CC}]\}$ denote the 3-channel junctions of the CC view mammography scan and corresponding mass and MC segmentations (from our proposed segmentation model), $\mathcal{X}_C^{MLO} = \{[\boldsymbol{x}_{MLO}, \boldsymbol{ma}_{MLO}, \boldsymbol{mc}_{MLO}]\}$ the same joint channel junctions relative to the same exam MLO view, and $\mathcal{Y}_C = \{0,1\}^3$ the proposed reformulated 3-class labelling.

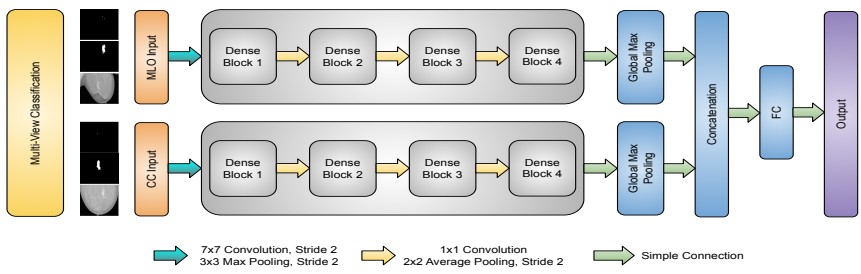

Figure 3: Classification Model Architecture.

## 3. Experimental Setup

This section will now describe the implementation details, and used dataset to evaluate our proposed fully-automated pipeline.

### 3.1. Implementation Details

To train the segmentation model, a combined loss function of both Dice and Cross Entropy Losses was used. This was inspired by the works (Sun et al., 2019; Zhu et al., 2019), which state that solely using Dice Loss can lead to unstable optimization. Furthermore, the presence of a Dice Loss component forces the model to detect the positive class (lesions). Specifically, the loss used for the segmentation model was

$$L_{Combined}(y_t, \hat{y}_t) = L_{Dice}(y_t, \hat{y}_t) + L_{BCE}(y_t, \hat{y}_t), \tag{1}$$

where $L_{Dice}(y_t, \hat{y}_t)$ and $L_{BCE}(y_t, \hat{y}_t)$ denote the dice and cross entropy, respectively.

To improve generalization, weight decay and data augmentation through random rotation of angle $\theta$ was used. L2 weight regularization led to stabler generalization in comparison to L1 regularization, while data augmentation helped twofold by increasing training size and improving segmentation. Exponential weight decay was used as follows:

$$\eta_p = \eta_0 \times \gamma^p, \tag{2}$$

where $\eta_p$ denotes the Learning Rate (LR) at training epoch $p$, $\eta_0$ is the initial value for the LR, and $\gamma$ the decay factor, respectively. To summarize, the hyper-parameters used were set as follows: the compression factor and growth rate (Huang et al., 2017) were kept at recommended values 0.5 and 32, respectively; starting LR set to $\eta_0 = 10^{-4}$; LR decay set to $\gamma = 0.96$, to follow the otherwise step decay proposed in (Sun et al., 2019); L2 weight regularization factor set to $10^{-5}$, rotation angle $\theta \in [-20°, 20°]$; batch size set to 8; and training was optimized with Adam using ($\beta_1 = 0.9, \beta_2 = 0.999, \epsilon = 10^{-7}$) for 100 epochs.

Regarding the classification model, training used cross-entropy as the loss function. L2 regularization also proved better than L1. Still facing sub-optimal generalization, dropout proved to match closer training and testing performance. When dealing with class imbalance, data augmentation through random rotation and flipping were tested. Furthermore, class weighting (Provost, 2000) was also experimented on. The latter delivered better results, even when using data augmentation generating balanced batches. The differences in hyper-parameters to the classification training were: Starting LR set to $10^{-3}$; LR decay fixed to 0.95; batch size set to 32; and dropout set to 0.2.

Training was executed under the GPUs publicly available by Google Colab (Google). Models were implemented in Python, assisted with libraries such as Keras (Chollet et al., 2015). Transfer Learning was employed for the DenseNet-121-BC core of both models, pre-training on Imagenet (Deng et al., 2009). Otherwise, weights were set with a normal distribution, following Xavier initialization (Glorot and Bengio, 2010).

### 3.2. Dataset and Pre-Processing

The INbreast dataset consists of 410 annotated scans with precise segmentation maps of masses and Micro-Calcification (MC) lesions. These scans have an imbalanced distribution of 6 classes, corresponding to the Breast Imaging Report and Data System Score (BIRADS) (Balleyguier et al., 2007), which were reformulated into 3 classes, due to low class frequency: Normal (BIRADS = 1); Benign (BIRADS $\in$ {2,3}); and Malignant (BIRADS $\in$ {4,5,6}). Both distributions are shown in Figure 4.

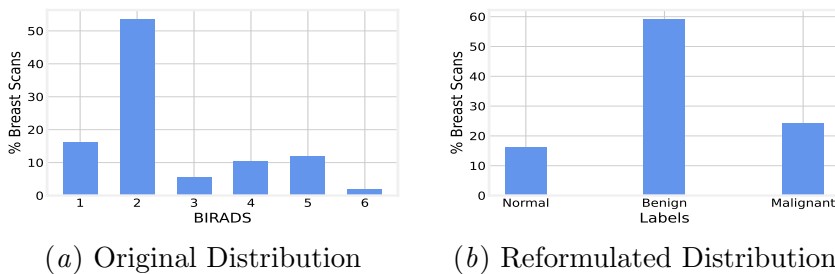

($a$) Original Distribution  ($b$) Reformulated Distribution

Figure 4: Original and Reformulated INbreast Dataset Class Distributions.

Besides being public, a reason for adopting this dataset lies in its limited sample size. Having a small training dataset constitutes a major roadblock for DL based techniques. Herein, we aim to show that our proposal overcomes this limitation with the aid of techniques like Transfer Learning (TL) on the Imagenet dataset (Deng et al., 2009), as in (Carneiro et al., 2017). The data dependence was further relieved by data augmentation and class weighting in lesion segmentation and classification, respectively.

The pre-processing followed the same methodology as in (Carneiro et al., 2017), where mammograms were enhanced via local contrast normalization, followed by Otsu's segmentation (Otsu, 1979) to select a tight bounding box containing the breast region, and flipping, so that the pectoral muscle was always located on the right-hand side. To relieve computational and memory costs, images were then downsized via bi-cubic interpolation to 264x264 resolutions. The data then was normalized to have zero-mean and unit standard deviation.

Given the small size of the INbreast dataset, validation through stratified k-fold cross-validation (with k = 5) was used, and metrics were averaged between splits to provide a more meaningful model evaluation.

### 3.3. Evaluation

The performance of the segmentations was evaluated using Dice Coefficient, and relative area difference, $\Delta A$, given by

$$\Delta A = \frac{|A_{GT} - A_{Pred}|}{A_{GT}} = \frac{|(TP + FN) - (TP + FP)|}{TP + FN}, \tag{3}$$

where $A_{GT}$ and $A_{Pred}$ represent the ground truth and predicted lesion area, respectively.

The proposed classification model was evaluated using metrics of Lesion Classification AUC (2-Class AUC of "Benign vs Malignant", where it is assumed that all cases contain at least one mass or MC), Breast Screening AUC (2-Class AUC of "Normal/Benign" vs "Malignant"), Mean AUC (2-Class AUC mean, where each class is considered positive and the remainder negative, sequentially), Sensitivity, and Specificity.

## 4. Results

### 4.1. Segmentation Results

Our proposed model achieves a dice performance $(0.71 \pm 0.08)$, and $\Delta A$ score of $(0.35 \pm 0.11)$ in mass segmentation. These results are comparable to the state of the art (Sun et al., 2019), which reports a dice coefficient of $(0.79 \pm 0.06)$ and $\Delta A$ of $(0.38 \pm 0.15)$. Regarding MC segmentation, our proposed model achieves good $\Delta A$ performance $(0.26 \pm 0.04)$, in comparison to the mass segmentation. No additional suitable comparison could be made, due to lack of records concerning similar approaches (i.e. single-stage model segmentation) on the INbreast dataset. The segmentation performance is crucial to the objective of this work, which lies in the fully-automated classification of mammography images (see Section 4.2).

Some samples of predicted lesions by our proposed model are depicted in Figure 5. It is to note that, while the representation overlaps mass and MC lesions and joins both views for each exam, the segmentations were separately produced by the respective Single-View model for each lesion type. The outlines in red and orange depict ground truth masses and MCs, while blue and cyan outline the segmentation prediction of masses and MCs.

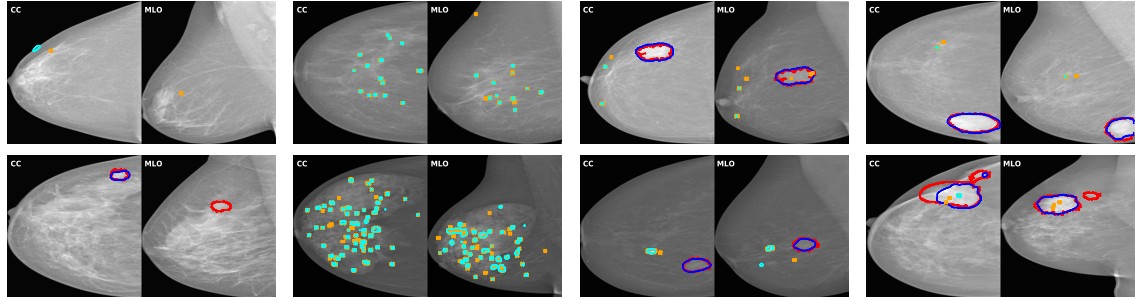

Figure 5: Proposed Model Lesion Segmentation of Test Samples.

As a preliminary conclusion to these results, we can note that TL helped model initialization, leading to better overall performance. However, while the TL improvements using ImageNet were substantial, they did not significantly surpass the performance of (Sun et al., 2019), which pre-trained on CBIS-DDSM (Lee et al., 2017). This goes according to the fact that not only the INbreast and CBIS-DDSM datasets share similar domains (mammography images), but also due to the experiment in (Sun et al., 2019) sharing pre-trained and fine-tuned model tasks (mass segmentation). Furthermore, their experiment only trained and tested on INbreast cases containing mass lesions, whereas our proposed model trained on the entire dataset, having no bias towards the presence of lesions.

### 4.2. Classification Results

Our proposed fully-automated classification pipeline achieves better than state of the art performance (namely, a mean AUC of $0.79 \pm 0.06$). Comparisons to state of the art are presented in Table 1, where the column 'LC AUC' and 'BS AUC' denote the Lesion Classification AUC and Breast Screening AUC metrics found in (Carneiro et al., 2017), respectively.

Table 1: Fully-Automated Classification Comparison to State of the Art.

| Method | LC AUC | BS AUC | Mean AUC | Sensitivity | Specificity |
|---|---|---|---|---|---|
| Proposed | **0.86**±**0.07** | 0.85±**0.03** | **0.79**±**0.06** | 0.61±**0.04** | **0.80**±**0.03** |
| (Carneiro et al., 2017) | 0.78±0.09 | 0.86±0.09 | 0.72±0.10 | **0.66**±0.14 | 0.69±0.23 |
| (Dhungel et al., 2017) | - | 0.80±0.04 | - | - | - |
| (Zhu et al., 2017) | - | 0.86±**0.03** | - | - | - |

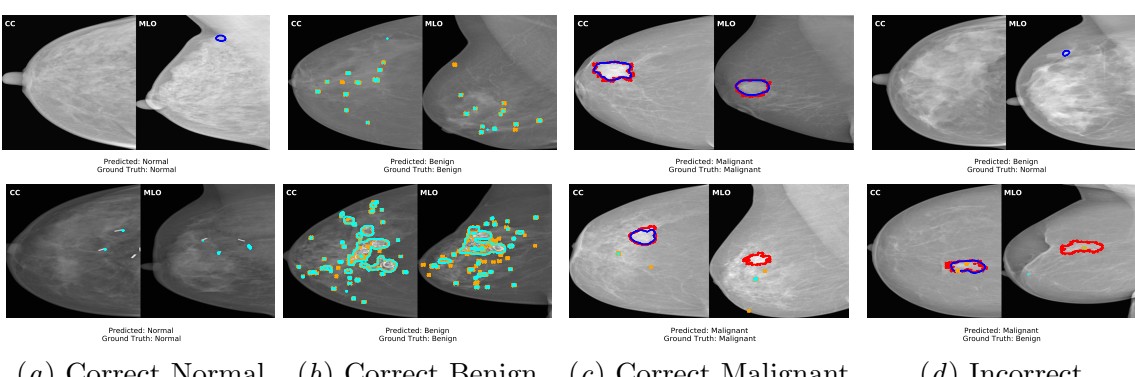

(*a*) Correct Normal    (*b*) Correct Benign    (*c*) Correct Malignant    (*d*) Incorrect

Figure 6: Proposed Model Correct (a-c) and Incorrect (d) Classification of Test Samples.

Some classifications of test cases by our proposed Multi-View model are depicted in Figure 6. The first three columns present correct classifications of each class, while the last column presents incorrect classifications. We remind that the model input leveraged auxiliary information channels containing lesion masks provided by our proposed segmentation models. Again, the colours red and orange denote ground truth masses and MCs, while blue and cyan regard the input segmentation masses and MCs, accordingly.

### 4.3. Ablation Studies

In order to further validate the proposed fully-automated approach, three ablation studies were conducted (see Table 2, where the column description follows Table 1):

1. An Image-Only Classifier evaluated the performance of a model analogous to the proposed classification model, but exclusively relying on mammography scan inputs (i.e. no lesion mask channels were added to the input);

2. A Single-View Classifier utilized a Single-View adaptation of the proposed model for classification (i.e. a model with only the top row of Figure 3) to verify the proposed Multi-View topology;

3. A Semi-Automated Classifier provided an upper bound to our proposed model by replacing the segmented lesion input of the fully-automated pipeline (see Figure 1) with the dataset ground-truth segmentations.

Table 2: Ablation Studies Vs. Proposed Model Performance

| Method | LC AUC | BS AUC | Mean AUC | Sensitivity | Specificity |
|---|---|---|---|---|---|
| Image-Only Classifier | 0.65±0.05 | 0.64±0.09 | 0.65±**0.05** | 0.50±0.09 | 0.75±0.04 |
| Single-View Classifier | 0.77±0.08 | 0.74±0.10 | 0.75±0.06 | 0.58±0.08 | 0.78±0.05 |
| Semi-Automated Classifier | **0.88**±**0.04** | **0.86**±0.05 | **0.86**±0.06 | **0.70**±0.08 | **0.84**±0.05 |
| Proposed | 0.86±0.07 | 0.85±**0.03** | 0.79±0.06 | 0.61±**0.04** | 0.80±**0.03** |

To summarize the preliminary conclusions, we should highlight that the inclusion of lesion masks substantially increased classification performance across all metrics versus the Image-Only Classifier, validating the information gained when using this technique. It is also worth noting that the presence of masses/MCs in the input does not discriminate between classes Benign and Malignant, as well as a 24% frequency of Normal cases containing at least one segmented mass or MC by our proposed model (eg. the upper exam of Figure 6(a)). Thus, a correct classification of samples indicates that our model is robust.

Furthermore, when faced against the Single-View Classifier, the Multi-View topology of our proposed model showed substantial increase in performance. This could be explained by the fact that the presence or absence of lesions detected in one view produce informative features relevant in the Multi-View classification (seen in the upper and lower exams of Figures 6(a) and 6(c), respectively). In contrast, Figure 6(d) presents two over-sensitive classifications of this phenomenon.

Moreover, the Semi-Automated Classifier serves as an the upper bound to this experiment by having ground truth segmentations in its input. The results are, as expected, higher than the fully-automated pipeline. An interesting observation can be made when comparing the results of our semi-automated classifier and the results of the semi-automated pipeline of (Carneiro et al., 2017) - LC AUC (0.94±0.05), BS AUC (0.91±0.08), Mean AUC (0.87 ± 0.08), Sensitivity (0.69 ± .28), and Specificity(0.92 ± 0.08). Since the classification performance is comparable, our proposed model improvements in Table 1 should lie in the provided segmented lesions. This seems to be an indication that indeed the lesion segmentations provided by our proposed segmentation model contributed to a more informative lesion interpretation in mammography.

## 5. Conclusions

In an attempt to assist practitioners with CAD in the field of mammography screening, radiologist performance has been shown to improve using novel DL model architectures. In this work such tools were developed for the task of both classification and segmentation of lesions in mammography exams. More concretely, this work proposed sophisticated models that were separately capable of labelling unseen exams according to their lesion magnitude and detecting possible suspicious lesions. Furthermore, this work proposed the use of an holistic lesion detection DL model to provide competitive fully-automated classification performance relative to the state of the art.

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
