# OpenReview forum: "Fully-Automated Multi-View Classification for Lesion Interpretation in Mammography"
_MIDL.io/2021/Conference — Submitted to MIDL 2021_

### Official Review · AnonReviewer4 · 2021-03-08

**Confidence:** 4
**Preliminary Rating:** 1

**Summary:**

This paper addressed the problem to classify and segment the lesions from in Mammography using multiple modalities. An adaptive attention U-Net and a dense-layer-based classification network are utilized and the experiments on INbreast dataset show a fair performance. Some illustrations showed the general performance.

**Strengths:**

1. The authors described their network structure clearly, and the paper is technically solid.
2. The experiments on a public dataset are provided, which makes it easier to reproduce the results from the authors' descriptions.

**Weaknesses:**

1. The paper is lack of novelty as it is more of a technical report of well-known network structures and tasks. The authors didn't provide much novel ideas in the paper.
2. The experimental results are only compared with the results published in 2017, which are unfair as more recent works may have much higher performance.

**Deanonymize Review:**

no

**Justification Of The Preliminary Rating:**

Although the authors tries to use the relatively new attention U-Net and dense layer based classification network, the paper is lack of novelty, and the results are not comparable with some of SOTA networks that are not listed and compared.

**Paper Type:**

validation/application paper

**Questions To Address In The Rebuttal:**

Please provide more comparisons with more recent SOTA works.

**Special Issue:**

no

---

### Official Review · AnonReviewer3 · 2021-03-09

**Confidence:** 4
**Preliminary Rating:** 1

**Summary:**

Given the extreme importance in early detection of breast cancer, a compelling search for Computer-Aided Detection (CAD) techniques drove Deep Learning (DL) researchers to investigate potential mammography screening applications. This work proposes the use of sophisticated and recently proposed Convolutional Neural Networks (CNNs) for classification and segmentation of mass and Micro-Calcification (MC) lesions on the INbreast dataset via a fully-automated pipeline.

**Strengths:**

The results seem reasonable: The classification stage is performed via a Dense Multi-View model, receiving an enriched input with the previous predicted segmentations, achieving performance on par with state of the art for fully-automated classi cation of breast screening exams (Normal, Benign, Malignant), achieving a 3-Class Mean AUC of (0:79 +- 0:06).

**Weaknesses:**

The novelty of the network is very limited: This work proposes the use of sophisticated and recently proposed Convolutional Neural Networks (CNNs) for classification and segmentation of mass and Micro-Calcification (MC) lesions on the INbreast dataset via a novel fully-automated pipeline. For segmentation, an Attention Dense U-Net model is used to provide segmentations for masses for MCs. I didn't see anything new here.

**Deanonymize Review:**

no

**Justification Of The Preliminary Rating:**

The novelty of the network is very limited: This work proposes the use of sophisticated and recently proposed Convolutional Neural Networks (CNNs) for classification and segmentation of mass and Micro-Calcification (MC) lesions on the INbreast dataset via a novel fully-automated pipeline. For segmentation, an Attention Dense U-Net model is used to provide segmentations for masses for MCs. I didn't see anything new here.

**Paper Type:**

methodological development

**Special Issue:**

no

---

### Official Review · AnonReviewer2 · 2021-03-09

**Confidence:** 5
**Preliminary Rating:** 1

**Summary:**

This work proposes the use of sophisticated and recently proposed Convolutional Neural Networks (CNNs) for classiﬁcation and segmentation of mass and Micro-Calciﬁcation (MC) lesions on a publicly available dataset of small size. Authors use combination of  an attention Dense U-Net and a Dense Multi-View model to predict classiﬁcation of breast screening exams (Normal, Benign, Malignant).

**Strengths:**

+ simple approach to use mass and micro-calcification segmentation to provide attention for malignancy classification.
+ Use of open and publicly available dataset
+  Good description of training procedure for reproducibility.

**Weaknesses:**

+  Very weak research into literature. Authors need to cite a lot more recent works in this field, even on same datasets and compare their results to newer publications. For example: [Shen, L., Margolies, L.R., Rothstein, J.H. et al. Deep Learning to Improve Breast Cancer Detection on Screening Mammography. Sci Rep 9, 12495 (2019)](https://doi.org/10.1038/s41598-019-48995-4) and [Y. Shen et al. An interpretable classifier for high-resolution breast cancer screening images utilizing weakly supervised localization Medical Image Analysis Volume 68, February 2021, 101908](https://doi.org/10.1016/j.media.2020.101908) etc.
+ Given usage of almost 1/10th size of original images, its not clear what is the impact of such a drastic resizing on micro-calcifications.
+ Authors do not show any impact of small data size on the reported metrics.




**Deanonymize Review:**

no

**Detailed Comments:**

+ Some of the colors used in figure 3 are not clear. For example what is the meaning of dar gray, dark green and dark yellow arrows?
+ Language of the paper is more like a project report rather than of publication quality.


**Justification Of The Preliminary Rating:**

Given the quality of writing that is more like a project report, lack of proper comparisons with literature (see points under weaknesses)................................................................

**Paper Type:**

validation/application paper

**Special Issue:**

no

---

### Official Review · AnonReviewer1 · 2021-03-09

**Confidence:** 5
**Preliminary Rating:** 1

**Summary:**

The authors propose a two-stage approach for classifying lesions in mammograms. First, an Attention Dense U-Net is applied for the two mammographic views (MLO and CC) to segment micro-calcifications. These segmentations are fed together with the image to a classification network. The selling point of this paper is a simplified, more holistic architecture compared to related work.


**Strengths:**

The authors offer insightful ablation studies that reveal the importance of segmentation maps, multiple views, and the influence of the predicted vs. ground-truth segmentations.

The clinical application is relevant and the complexity of the proposed model is moderate.



**Weaknesses:**

The contributions of this paper remain unclear. It is stated that the Attention Dense U-Net (proposed by Li et al.) was adapted in Figure 2a. However, what are the adaptations?

It is emphasized that the proposed network is "holistic". Why is it not trained end-to-end using an auxiliary loss for the segmentation layer?

The performance difference compared to the work of Sun et al. is quite large (0.71 vs. 0.79). How can this be explained?

A couple of model decisions could be further quantified, e.g., what is the influence of transfer learning for the classification network? How crucial are data augmentations?

**Deanonymize Review:**

no

**Detailed Comments:**

The Figure reference in Section 2.2 needs to be corrected (from Fig. 2 to Fig. 3).


**Justification Of The Preliminary Rating:**

It is unclear what the contributions of this paper are. Most of the model components have been adopted from previous work, e.g., the Attention Dense U-Net and other design decisions have not been carefully studied.

The reported results are not competing with the state-of-the-art but it is not discussed why.

**Paper Type:**

validation/application paper

**Questions To Address In The Rebuttal:**

What are the contributions of this paper?

What is the influence of additional model choices (classification, data augmentation)?



**Special Issue:**

no

---

### Meta-Review · Area_Chair1 · 2021-03-24

**Recommendation:** Reject

**Metareview:**

All reviewers suggest rejection, mainly due to lack of novelty and methodological developments and limited assessment of related work. The authors have also decided not to rebut the reviewer comments and as such I go with the initial rating of the reviewers.

**Paper Type:**

validation/application paper

---

### Decision · Program_Chairs · 2021-03-31

Reject